# The Effects of Dietary Nutrition Education on Weight and Health Biomarkers in Breast Cancer Survivors

**DOI:** 10.3390/medsci5020012

**Published:** 2017-06-02

**Authors:** Andrea Braakhuis, Peta Campion, Karen Bishop

**Affiliations:** 1Discipline of Nutrition and Dietetics, FM & HS, University of Auckland, Private Bag 92019, Auckland 1142, New Zealand; pcam131@aucklanduni.ac.nz; 2Auckland Cancer Society Research Center, FM & HS, University of Auckland, Private Bag 92019, Auckland 1142, New Zealand; k.bishop@auckland.ac.nz

**Keywords:** Mediterranean diet, breast cancer, low-fat diet, weight, nutrition education

## Abstract

Weight gain after breast cancer diagnosis portends a poorer prognosis, and the majority of sufferers appear to gain weight. Metabolic syndrome is a common co-condition with breast cancer. The Mediterranean diet has been used to reduce excess weight, metabolic syndrome, and to improve the inflammatory profile, and therefore may offer the breast cancer survivor specific benefits over and above the currently recommended nutrition guidelines to eat a low fat, healthy diet. The aim of this randomised controlled trial was to investigate whether a Mediterranean (MD) or low-fat diet (LF) reduce weight and general health in survivors of stage 1–3 breast cancer through a six-month, six-session education package to support dietary change. A control dietary arm received no intervention. Outcome measures for weight, body mass index (BMI), waist circumference, blood lipids, blood glucose, dietary adherence, 3-day food diary, and PREDIMED questionnaire and quality of life were assessed. Both dietary intervention arms, on average, lost weight over the course of the intervention, with significant (*p* < 0.05) decreases seen in BMI and waist circumference measurements. The control arm gained weight and significantly (*p* < 0.05) increased BMI and waist circumference measurements overall (1.10 ± 3.03 kg, 0.40 ± 1.65 kg/m^2^, and 1.94 ± 2.94 cm respectively). Positive trends in blood biomarkers were observed for the intervention arms. Dietary adherence was sufficient. Nutritional education and group support appears to exert beneficial effects on health in breast cancer survivors, of lesser importance is the type of diet that forms the basis of the education.

## 1. Introduction

On the basis of observational studies, women with breast cancer who are overweight or gain weight after diagnosis are found to be at greater risk for breast cancer recurrence and death compared with lighter women. Obesity is also associated with hormonal profiles likely to stimulate breast cancer growth [1]. Although definitive weight loss intervention trials in breast cancer patients remain to be conducted, the current evidence relating increased body weight to adverse breast cancer outcome and the documented favourable effects of weight loss on clinical outcome in other comorbid conditions support the consideration of programs for weight loss in breast cancer patients. Evidence exists for a connection between obesity, metabolic syndrome, and poor outcomes in breast cancer patients [1]. Metabolic syndrome is also known as insulin insensitivity syndrome, and is defined as central obesity together with two of the following risk factors: elevated glucose, insulin resistance, elevated triglycerides, reduced high density lipoproteins (HDLs), and/or hypertension. Breast cancer patients with metabolic syndrome undergoing chemotherapy are found to respond poorly to treatment [1], although the causal mechanisms are not known. Patients with high blood glucose levels are also noted to have an increased rate of breast cancer progression [1]. 

Due to indications of links between better survival after breast cancer, both the American Institute for Cancer Research (AICR) and the American Cancer Society (ACS) currently suggest the maintenance of a healthy body weight and the achievement of a dietary pattern rich in vegetables, fruits, and whole grains for long-term disease-free living [2]. The current dietary recommendations that advocate for a healthy, low-fat diet may be ignoring some of the recent research on the health benefits of the Mediterranean diet (MD). Dietary polyphenols found in a MD have been reported to reduce inflammation and cancer recurrence through various mechanisms, including direct antioxidant activity or antioxidant gene expression, inhibiting cancer cell proliferation, cytokines and endotoxin-mediated kinases, and transcription factors involved in cancer progression or increasing histone deacetylase activity [3]. It is possible that the MD confers additional benefit to the breast cancer survivor compared to the currently recommended low-fat healthy eating. 

The MD is rich in olive polyphenols, which have the potential to inhibit the progression of breast cancer. Olive polyphenols have demonstrated the ability to inhibit the proliferation of several cancer cell lines, including breast [4]. Particular olive polyphenols—oleuropein and hydroxytyrosol—have consistently been reported to discriminate between cancer and normal cells, inhibiting proliferation and inducing apoptosis only in cancer cells [5]. The manipulated pathways and signalling cascades include the nuclear factor kB (NF-kB) inflammatory response and oxidative stress pathways, the polyphenols may also act as phytoestrogen mimics. Due to the similar structure of the olive polyphenols to estrogen, these have been hypothesised to interact with estrogen receptors, thereby reducing the progression of hormone-related cancers. Evidence for the protective effect of olive polyphenols for cancer in humans remains anecdotal, and clinical trials are required.

The nutrition guidelines for breast cancer sufferers are typically generic, and there is considerable confusion as to the role of diet in post-diagnosis survival. Dietary intervention and behaviour change is a potentially cost-effective alternative for preventing the large burden of healthcare associated with breast cancer treatment. The aim of the study was to investigate whether the MD had advantages over the currently adopted low-fat healthy eating recommendations, and whether any dietary intervention was more beneficial than no dietary support.

## 2. Materials and Methods

The study was offered to local (Auckland, New Zealand) breast cancer survivors of any ethnicity. A six-month, three-arm, parallel-randomised control trial with baseline testing and post-intervention testing was conducted. Specifically, it focused on the effects that dietary changes bring to the body in terms of weight, waist circumference, body mass index (BMI), and blood lipid and glucose profiles. The study also explored the effects that nutrition education had on the uptake of dietary changes and the outcome results.

### 2.1. Participants

Trial participants were post-menopausal women aged 50 years and over who had previously been diagnosed and treated for stage 1–3 breast cancer, confirmed by individual medical data. Informed consent was obtained from all individual participants included in the study. The study was approved by the Health and Disability Ethics Committee, New Zealand, reference, 15/STH/184/AM01. Participants were eligible for the trial if they were three or more months following active treatment (chemotherapy and surgery), were no more than 3 years after active and hormone treatment, and had a BMI > 25. Present or past hormonal therapy was not an exclusion criterion. The participants were excluded from the trial if they were on anti-inflammatory medication, drank more than two standard alcoholic beverages per day, smoked tobacco, or were diagnosed with poorly controlled diabetes mellitus. 

Potential participants (180) contacted the research team for information. Of these, 75 showed further interest, with 25 of these excluded on the basis of being too young, having advanced stage cancer, or unable to attend the education sessions. Fifty were recruited and completed baseline testing.

### 2.2. Dietary Arms

Participants were randomly block allocated to one of three sets. Set 1 was deemed Arm 1: the MD, set 2 was titled Arm 2: the healthy, low-fat diet (LF), and set 3 became Arm 3: the no-treatment control. Consequently, 17 participants were assigned to Arm 1, 16 to Arm 2, and 17 to Arm 3.

Diets were ad libitum. The MD arm were provided with olive leaf extract to trial; while the LF arm were provided complimentary rice bran oil. The MD diet was based on the main dietary principles of the Mediterranean region and consisted of increasing the intake of olive oil, vegetables, fruit, legumes, fish, nuts, onions, leeks, tomatoes, and garlic while limiting the intake of red meat, butter, margarine, cream, carbonated drinks and sweets, chocolate and baking. The LF diet was based on the New Zealand Ministry of Health Eating and Activity Guidelines for Adults [6].

### 2.3. Intervention Education Sessions

Dietary Arms 1 and 2 each received six group nutrition and lifestyle education sessions from the same educator. The no-treatment control group did not receive any dietary guidance or intervention over the course of the study. Education sessions occurred once monthly, with a summary newsletter sent two weeks thereafter. Attendance to the scheduled group education sessions was monitored via roll call at each session. Each intervention group received a total of six education sessions and six newsletters. The education topics are outlined here, numbered according to education session:Introduction to group members, the assigned diet, and the concept of lifestyle change including dietary, activity, and social aspectsDiscussion of barriers to achieving dietary, activity, and lifestyle goals, and how the environment can affect weight and lifestyle choicesPractical exercise on calorie definition and estimating portion sizes including education on completing food diariesDiscussion around stress management and fatigue in terms of eatingPractical exercise in correctly reading food labelsIntroduction of weight loss tips to the participantsDiscussion of tips for shopping successfully to maintain new dietary habitsSession was conducted via email to save participant travel. It involved a recap of the assigned diet and education of correct portion sizes and maintaining portion controlDiscussion between the participants on their experience with the study and the changes they had made, with focus given to maintenance, staying in control, and avoiding relapse


### 2.4. Dietary Assessment

#### 2.4.1. PREDIMED Questionnaire

Participants completed PREDIMED questionnaires at baseline and post-intervention. The 14-item questionnaire was the primary measure used in this study to appraise adherence of participants to the MD. The characteristics of participants were compared according to three categories of adherence to the Mediterranean diet (≤5 (low), 6–9 (medium), and ≥10 (high) points of the 14-item questionnaire). Each question was scored 0 or 1. If the condition of a question was not met, 0 points were recorded for the category. The final PREDIMED score ranged from 0 to 14.

#### 2.4.2. Three-Day Food Diary

Participants were required to complete one three-day food diary during the intervention period. Verbal (face-to-face and/or telephonic) instructions were provided to the intervention groups, written instructions and phone support were provided to the control group. Collected data reflected a combination of actual (measured or weighed) and estimated food consumption. The diaries were assessed and analysed for total servings across a range of food groups (grains and grain products, milk and dairy products, chicken, red meats and its products, eggs, fish and seafood, fruit, vegetables, nuts or seeds, legumes and beans, sweets, baking and chocolate, sugar, honey, olive oil, other fats and oils, coffee, alcohol, sweet drinks, and tea). Standard serving sizes were used, according to the New Zealand Ministry of Health Nutrition guidelines [6].

### 2.5. Quality of Life. FACT-B Questionnaire

Participants were asked to complete a functional assessment of cancer therapy-breast (FACT-B) questionnaire at baseline and post-intervention to assess the effects of group education. This questionnaire is a self-reporting tool designed to measure multi-dimensional quality of life in patients with breast cancer, and is split into six subscales, namely physical well-being (PWB), social/family well-being (SWB), emotional well-being (EWB), functional well-being (FWB), relationship with doctor (RWD), and the additional concerns for breast cancer (BCS). A higher score indicates a higher health-related quality of life in the participant.

### 2.6. Blood Tests

The glycosylated haemoglobin percentage (HbA1c %) and lipid panels were conducted on whole blood immediately following blood sample collection in EDTA vacutainers, with remaining sample aliquots frozen at −80 degrees Celsius for later analysis using the Afinion AS100 Test Cartridge (Waltham, MA, USA).

### 2.7. Anthropometry

All measurements were conducted using the International Society for Anthropometry and Kinanthropometry (ISAK) protocol by a Level 3 ISAK Certified Practitioner. Weight (kg) was taken on calibrated load scales (Advasco Scales Ltd., Auckland, New Zealand) without shoes or extraneous clothing. Height (m) was taken as a freestanding height measurement on a stadiometer (University of Auckland, Grafton Campus) without shoes. BMI was calculated using the equation: weight (kg)/height (m)^2^. The weight measurements were taken at two time-points—at baseline and at post-intervention—using the same scales. Height was recorded at baseline. BMI was calculated at the same time as each weight measurement, using the single height measurement as the denominator. Waist circumference was measured twice—once at baseline and once post-intervention—using Lufkin steel tape. All measurements were taken by the same individual.

### 2.8. Outcome Measures

The primary outcome measures for this study included: change in body weight (kg), BMI (kg/m^2^), and waist circumference (cm).

The secondary outcome measures included: change in lipids (total cholesterol, low-density lipoprotein (LDL) cholesterol, HDL cholesterol, non-HDL cholesterol, total cholesterol:HDL cholesterol ratio, and triglycerides), HbA1c %, adherence to a Mediterranean eating pattern (PREDIMED questionnaire), quality of life score (FACT-B questionnaire), group education attendance, and three-day food diary intake.

### 2.9. Data Analysis

SPSS Version 23.0 software (IBM Corp., Armonk, NY, USA) was used for all statistical analyses. Data was assessed for normality using the D’Agostine & Pearson normality test, and data was normally distributed. The study compared the means and standard deviations (SD) or 95% confidence intervals (CI) of outcome indexes (weight, BMI, waist circumference, blood lipid levels, PREDIMED and FACT-B questionnaire data) across the three dietary arms. Associations between these baseline and post-intervention measurements were evaluated using one-way repeated measures analysis of variance (ANOVA). Significance was defined as *p* < 0.05. Statistical analyses excluded missing and undetermined data. Tukey honest significant difference (HSD) post-hoc testing was performed if there were statistically significant differences in the results with the aim of determining where those differences lay.

## 3. Results

The final analysis was conducted on those participants who completed the study (MD *n* = 15, LF *n* = 12, control group *n* = 13). Baseline characteristics are shown in Table 1. The mean BMI across all dietary arms showed that the participants were categorised as being overweight according to the BMI criteria (<18.5 underweight, 18.5–24.9 normal, 25–29.9 overweight, and >30 obese). The vast majority of participants (68%) identified as being of New Zealand European ethnicity; all other ethnicities are listed in Table 1. All other variables, including weight, waist circumference, HbA1c %, total cholesterol, HDL and LDL cholesterols, triglycerides, non-HDL cholesterol, and total cholesterol:HDL cholesterol ratio levels were similarly matched across the three dietary arms. There were no deaths over the course of this study.

Greater weight loss was observed among MD participants, compared to the LF participants, whilst the control arm participants on average gained weight (Table 2). The range of weight change was greatest in the Mediterranean group. 

Following analysis using a one-way ANOVA to compare means, the BMI and waist circumference variables experienced significant differences between groups (*p* = 0.047 and *p* = 0.045, respectively). The control group saw a small increase in score (Table 3). Significant differences were observed between groups in the post-intervention PREDIMED score.

No significant differences were observed between groups both pre- and post-intervention in the FACT-B quality of life questionnaire scores.

## 4. Discussion

The results of this study demonstrate that the adoption of an MD induces weight loss amongst participants and decreases BMI and waist circumference towards more desirable measurements. The LF dietary arm also lost weight; however, the changes were less marked. Conversely, the control participants gained weight, and increased waist circumference and BMI measurements, thus suggesting that dietary education and support for dietary change elicit these findings. The control group gained on average 1.10 kg per participant, suggesting that significant input is required to even maintain weight. As such, dietary educational input for healthy weight attainment is needed in the breast cancer survivor population. Participants demonstrated that they were able to adhere to either intervention.

At present, a diet low in saturated fat and high in fruit and vegetables is advocated to those with cancer, along with maintaining a healthy body weight. Previous research has demonstrated effective weight loss and improved inflammatory outcomes in both cardiac and diabetic populations [7]. Breast cancer patients typically gain weight post diagnosis, which is associated with a worse prognosis. As such, the MD may assist in weight loss and improve the inflammatory profile to a greater extent than the currently recommended healthy, low-fat diet. The MD is a plant-based dietary pattern characterized by a high intake of olive oil, legumes, whole grains, fruit, vegetables, nuts, seeds, and fish, and is rich in dietary polyphenols. The diet has been linked to a decreased risk of developing breast cancer [8]. Little research has been conducted on those already diagnosed with breast cancer.

There was a tendency for desirable shifts in total cholesterol, LDL, HDL, and total:HDL cholesterols from baseline to the final measures taken after the intervention in the dietary arms that received group education; however, the changes observed were not statistically significant. Those participants asked to follow an MD diet decreased the average total cholesterol the most, but the participants following an LF diet showed more remarkable desired changes in HDL, LDL, non-HDL, and total:HDL cholesterols, and triglyceride levels. However, the post-intervention measurements still fall within recommended parameters. 

Additionally, the control group showed no change in their total cholesterol or LDL cholesterol. The results regarding HbA1c %—a longer-term measurement of blood glucose levels—are less clear, and a true pattern cannot be seen from this study; however, the measurements were deemed to be within the normal range [9].

Follow-up studies should continue to include a no-treatment control group, but define the intervention to a particular histology and stage of cancer given the heterogeneous nature of the disease.

## 5. Conclusions

Nutritional education and group support appears to be beneficial to breast cancer survivors, and of lesser importance is the type of diet that forms the basis of the education. With regards to breast cancer, weight is an important factor worthy of intervention and monitoring past the stage of active treatment. Women would benefit from regular assessment, routine follow-up visits until treatment cessation, complimented with regular dietitian and dietary follow-up post-treatment.

## Figures and Tables

**Table 1 medsci-05-00012-t001:** Baseline characteristics of post-menopausal female trial participants by intervention groups.

Baseline Characteristics of Participants According to Study Group °			
Characteristic	Mediterranean Diet (N = 17)	Low-fat Diet (N = 16)	Control Diet (N = 17)
Age—years.	54.71 ± 6.20	55.19 ± 8.32	56.71 ± 6.66
Cancer grade—no. (%)			
1	4 (24)	3 (19)	5 (29)
2	8 (47)	5 (31)	6 (35)
3	5 (29)	7 (44)	6 (35)
Not Specified	0 (0)	1 (6)	0 (0)
Ethnicity—no. (%)			
NZ European	13 (76)	11 (69)	10 (59)
Maori	1 (6)	0 (0)	1 (6)
Dutch	0 (0)	1 (6)	0 (0)
NZ/Samoan/German	0 (0)	0 (0)	1 (6)
Chinese	0 (0)	1 (6)	1 (6)
American	0 (0)	0 (0)	1 (6)
South East Asian	0 (0)	0 (0)	1 (6)
Korean	0 (0)	0 (0)	1 (6)
Cook Island Maori	0 (0)	0 (0)	1 (6)
English	0 (0)	1 (6)	0 (0)
Australian	1 (6)	0 (0)	0 (0)
Indian-Mauritian	1 (6)	0 (0)	0 (0)
Not Specified	1 (6)	2 (12)	0 (0)
Smoker—no. (%)			
Yes	0 (0)	0 (0)	0 (0)
No	17 (100)	14 (88)	17 (100)
Not Specified	0 (0)	2 (12)	0 (0)
Type 2 Diabetes—no. (%)			
Yes	1 (6)	0 (0)	0 (0)
No	16 (94)	14 (88)	17 (100)
Not Specified	0 (0)	2 (12)	0 (0)
Hormone Replacement Therapy—no. (%)			
Yes	6 (35)	4 (25)	5 (29)
No	10 (59)	12 (75)	10 (59)
Not Specified	1 (6)	0 (0)	2 (12)
BMI ^◊^	29.31 ± 5.62	26.87 ± 4.99	27.51 ± 5.93
<25—no. (%)	4 (24)	6 (38)	6 (35)
25–30—no. (%)	3 (18)	4 (25)	5 (29)
>30—no. (%)	10 (59)	5 (31)	5 (29)
Weight—kg	80.62 ± 16.86	71.94 ± 14.69	72.17 ± 14.44
Waist Circumference—cm	89.35 ± 11.41	84.22 ± 11.05	87.32 ± 11.40
Height—cm	165.18 ± 7.58	165.74 ± 5.4	162.45 ± 7.20
HbA1c—%	5.6 ± 0.45	5.58 ± 0.36	5.5 ± 0.63
Total Cholesterol—mmol/L	5.40 ± 0.81	5.01 ± 1.05	5.67 ± 1.05
HDL—mmol/L	1.80 ± 0.47	1.66 ± 0.44	1.60 ± 0.51
LDL—mmol/L	3.04 ± 0.63	2.76 ± 0.72	3.18 ± 0.88
Total cholesterol:HDL cholesterol—mmol/L	3.21 ± 1.00	3.41 ± 1.13	3.72 ± 1.25
Triglycerides—mmol/L	1.29 ± 0.92	1.41 ± 0.93	1.67 ± 0.86
Non-HDL—mmol/L	3.58 ± 1.01	3.68 ± 1.28	3.87 ± 1.00

° Data are presented as means ± SD. ^◊^ Body-mass index is the weight in kilograms divided by the square of the height in metres. BMI: body mass index; HbA1c: glycosylated haemoglobin; HDL: high-density lipoprotein; LDL: low-density lipoprotein.

**Table 2 medsci-05-00012-t002:** Changes in measurements by average total loss per participant by dietary group following 6-month intervention period.

	Mediterranean Diet		Low-Fat Diet		Control	
	Mean	95% CI °	Mean	95% CI °	Mean	95% CI °
**BMI** ^◊,^^^,^*	−1.02 **	−2.14 to 0.08	−0.56	−1.16 to 0.05	0.58 **	−0.4 to 1.55
**Weight—**kg ^	−1.61	−4.0 to 0.84	−1.35	−2.8 to 0.17	1.10	−0.73 to 2.92
**Waist Circumference—**cm *	−1.40	−3.5 to 0.69	−1.31	−3.6 to 0.98	1.94	−0.08 to 3.5
**HbA1c—**%	−0.31	−0.94 to 0.34	−0.13	−0.44 to 0.17	−0.12	−0.3 to 0.06
**Total Cholesterol—**mmol/L	−0.07	−0.44 to 0.32	0.05	−0.31 to 0.41	0.00	−0.33 to 0.42
**HDL—**mmol/L	0.15	0.00 to 0.34	0.19 ***	0.05 to 0.34	0.25 ***	0.03 to 0.27
**LDL—**mmol/L	−0.18	−0.5 to 0.13	−0.19	−0.46 to 0.06	0.00	−0.2 to 0.24
**Triglycerides—**mmol/L	0.03	−0.28 to 0.33	0.12	−0.08 to 0.31	−0.13	−0.38 to 0.12
**Non-HDL—**mmol/L	−0.17	−0.43 to 0.15	−0.58	−0.31 to 0.30	−0.04	−0.33 to 0.25

° Data presented as mean change score and 95% confidence interval (CI) of the pre-post change. ^◊^ Body-mass index is the weight in kilograms divided by the square of the height in metres; ^ Negative number represents weight was overall lost, positive number represents weight was overall gained; * Represents significance between groups (*p* < 0.05); ** Represents significance lies between the corresponding groups (*p* < 0.05); *** Represent significant difference pre to post (*p* < 0.05). BMI: body mass index; HbA1c: glycosylated haemoglobin; HDL: high-density lipoprotein; LDL: low-density lipoprotein.

**Table 3 medsci-05-00012-t003:** Average PREDIMED questionnaire total score for each dietary arm pre- and post-intervention.

	One-Way ANOVA *p*-Value	MD°	LF°	Control°
Baseline score	0.265	6.35 ± 2.81	5.13 ± 2.00	6.29 ± 2.29
Post-intervention score	0.02 *	9.80 ± 1.52	7.25 ± 2.05	7.46 ± 2.22
Difference		+3.45	+2.12	+1.17

* Represents significance between groups (*p* < 0.05); ° Results presented as mean and SD. LF: low-fat diet; MD: Mediterranean diet.

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
