# Peer review of "The Effects of Dietary Nutrition Education on Weight and Health Biomarkers in Breast Cancer Survivors"

_medsci, 2017, doi:10.3390/medsci5020012_

Round 1
Reviewer 1 Report
The authors have proposed an interesting study design. The criteria for inclusion/exclusion of subjects in the study appear appropriate. However, the number of women recruited in the study (n = 50) is too small to allow definitive conclusions. For this reason, the results should be described as preliminary at best. In my opinion, the sample size should be increased to make (preliminary) data more robust.
Author Response
We have resubmitted the paper as a Short Communication to reflect the nature of the research and findings. We would comment that the sample size was sufficient to detect a significant change in waist circumference and BMI with dietary intervention and believe readers would be interested in the findings.The sample size was calculated to detect a change in anthropometric variables, which it did do. We believe as a Short Communication, this is worthy of publication.
Reviewer 2 Report
The authors address an issue of importance, namely evaluation of dietary nutrition education on weight and health biomarkers in breast cancer survivors but do so in an extraordinary small number of participants. The stated aim of the study was to “investigate whether the Mediterranean diet had advantages over the currently adopted low-fat healthy eating recommendations and whether any dietary intervention was more beneficial than no dietary support.” The study is well intended and the design would be sound if the sample size was substantially larger. The current results provide no reliable evaluation of the posed study question.
Even if the study involved a substantially larger higher number of participants, the valued to the science in this area would be limited. In December of 2016, the American Society of Clinical Oncology had a special issue in the Journal of Clinical Oncology devoted to “obesity and cancer.” One of those manuscripts provided a review of all randomized weight loss intervention trials in women with cancer. The conclusion in that manuscript and others in the series recommended that no further studies be conducted of questions testing dietary intervention in breast cancer survivors attempting to duplicate adherence findings seen in larger studies in women without breast cancer. The feasibility of implementing dietary interventions in women with early stage breast cancer entered shortly after completion of their initial therapy (including long-term endocrine therapy) has been established and no further studies need to be conducted.
If the study were to be reported, the report should be greatly condensed (by about 2/3), very little of the extensive discussion, for example, has relevance to the question asked and the data presented in the results section.
Author Response
Thank-you for the suggestion to review the Journal of Clinical Oncology Special Edition on Obesity and Cancer. I agree with the authors in the wasted effort in comparing weight loss in those with cancer to those who do not. In our study the control group are cancer patients.
Many of the studies mentioned in the meta-analysis are epidemiological studies, of which thousands of participants are needed. Our study was a randomised control study which requires far fewer subject numbers to detect a meaningful change. Our study was powered to detect a weight change of 3%. We did see a significant change in waist circumference and BMI.
We have redrafted the paper and request it is accepted as a Short Communication. We have rewritten the discussion to remove any unrelated conversation and shorten to reflect the publication type. We believe readers will be interested in the Mediterranean diet, given the potential for weight loss and other benefits.